

# Understanding rockfalls along the national road G318 in China: from
# source area identification to hazard probability simulation
**Lixia Chen[1*], Yu Zhao[1], Yuanyao Li[2], Lei Gui[3], Kunlong Yin[3], Dhruba Pikha Shrestha[4]**
[1] Institute of Geophysics and Geomatics, China University of Geosciences, Wuhan, 430074, China; lixiachen@cug.edu.cn;
cugzhaoyu@cug.edu.cn
[2] Institute of Geological Survey, China University of Geosciences, Wuhan, 430074, China; liyuanyao@cug.edu.cn
[3] Faculty of Engineering, China University of Geosciences, Wuhan, 430074, China; lei.gui@cug.edu.cn; yinkl@cug.edu.cn
[4] Department of Earth Systems Analysis, Faculty of Geo-Information Science and Earth Observation (ITC), University of Twente,
7500 AE Enschede, the Netherlands; d.b.p.shrestha@utwente.nl
* *Correspondence to:* lixiachen (lixiachen@cug.edu.cn);
**Abstract:** Rockfall hazard is frequent along the national road (G318) in west Hubei, China. To understand the distribution and
potential hazard prone to road G318, this study combines the result of a 3-years engineering geological investigation, statistical
modeling, and kinemics-based method to identify risky road sections. Rockfall source area cells are preliminarily identified by slope
angle threshold analysis and then selected by susceptibility method (Random Forest model and multivariate logistic regression
model) with the result of potential spatial probability. Temporal and size probabilities of source areas are separately calculated by
Poisson distribution and power-law distribution theory. To get the reaching probabilities and potential influence area of released
source areas, rockfall trajectory simulation was taken by Flow-R tools. In this process, an important parameter (reach angle) was
determined by back analysis and then validated by field investigation. Rockfall hazard probability is finally calculated by integrating
spatial, temporal, size probability, and reaching probabilities of source areas. The results show good fitness with the measurements
from field work. In the conditions of 5, 20, and 50 years return period, potential risky road sections are found out under two size
scenarios (larger than 1 000 m3, 10 000 m3). This research helps the local government to completely understand the rock falls from
source area existence and potential risk to roads.
**Keywords:** Rockfalls; Slope angle threshold; Random Forest model; Multivariate logistic regression model; Flow-R; Reach angle
**1. Introduction**
Rockfall is the main kind of geological hazard along roads in steep mountainous areas such as in the Himalayas, the Alps, in
the rocky mountains, in the Andes, etc. Also in China, rockfall is a common problem in mountainous areas. The national highway
G318 is the longest motorway (approx. 5476 km) in China, starting from Shanghai and passing through major cities such as Wuhan,
Chongqing, Lhasa and finally ending in Kodari, in China/Nepal border. The major part of the road (more than 70 percent) lies in
the mountainous areas. Because of the special geomorphological and geological set up, the road section(approx. 1302 km) in Hubei-
Chongqing has been exposed to frequent slope failures causing property damages and disruptions of traffic. In 2016, a family in a
pick-up van was lost because of a small volume but sudden rockfall along the    G318. Such kind of small size but high frequency
and intensity (e.g.velocity, energy) rockfalls are common in China, which can lead to human casualties and property loss (Whalley,
1984). To protect the people commuting on the roads, we have to understand where the rockfall source area is and its hazard level.
Once we know this, then suitable mitigation measures can be implemented.
In terms of source area identification, a large number of research results are available. Common methods for identifying the
source areas can be divided into two main types: geomorphic and geological. The geomorphic approach uses the slope angle
threshold (SAT) method to identify rockfall source area in which slope gradient map derived using digital elevation model (DEM)
is used (Jaboyedoff et al., 2003; Loye et al., 2009; Žabota et al., 2019; Liu et al., 2020). This is also the reason why some researchers
try to apply surveying techniques to identify source areas, such as Light Detection and Ranging (LiDAR) and terrestrial laser
scanners (TLS) (Fanos et al., 2020). The existing research results show that the critical SAT values vary from rockfall types and



study areas (e.g. >60◦ in Wieczorek et al.,1998, and Guzzetti et al.,2003; >45◦ in Jaboyedoff and Labiouse, 2003; >37◦ in Frattini et al., 2008; >48°in Matasci, Jaboyedoff et al., 2015). So, terrain data is an important basis for rockfall hazard assessment. The morphology-based method is simple in data-limited areas. But from the view-point of engineering geologists, other conditioning factors such as discontinuities and joint sets in rocks are very important (Guzzetti et al., 1998; Jaboyedoff et al., 2003; Frattini et al., 2008; Heckmann et al., 2016).

If data is available, more accurate source areas can be identified, such as empirical, statistical, or deterministic methods. An empirical expert evaluating system has been developed to access hazard susceptibility or probability, such as Rockfall Hazard Rating System (RHRS). It is a widely used method to identify the riskiest slopes on highways or coastal roads (Brawner et al., 1975; Pierson, 1993; Budetta, 2004; Li et al., 2009; Corominas et al., 2013). The system is gradually optimized by using optical remote sensing data from satellites or unmanned aerial vehicles (UAVs) (Oommen, 1984). Statistical methods, such as the Random Forest Model (RFM) and Multivariate Logistic Regression Model (MLRM) are applied in Geographic Information System (GIS), especially for large or small scale areas. RFM, a machine learning algorithm based on the concept of classification trees, is able to classify landslide hazard susceptibility (Chen et al., 2014; Messenzehl et al., 2017). MLRM is widely used to construct slope instability susceptibility models (Carrara, 1983; Chung et al., 1995) and has the advantage of being less demanding compared to other techniques such as discriminating analysis (Carrara, 1991; Baeza et al., 1996). RFM can achieve higher accuracy with the same data. However, different models result in different source area locations. Thus, it is important to know which model performs better in the area of interest.

Besides rockfall source area, we also need to know rock mass trajectory paths with resulting intensity (e.g. velocity or kinetic energy) and the area it can affect. To simulate the trajectories and energy, several 2D or 3D tools or software were developed for regional-scale or site-specific rock slopes, such as CADMA by Azzoni et al. (1995); CONEFALL by Jaboyedoff et al. (2003), Flow-R by Horton et al. (2013), STONE by Guzzetti et al., (2002), RAMMS by Leine et al. (2013), DDA by Zheng et al. (2014), Rockyfor3D (Dorren L.K.A., 2016). Between them, STONE, RAMMS, DDA, and Rockyfor3D can simulate three-dimensional collapse motion, but they all have a defect that all require extensive field investigation and experimental parameters. Flow-R is developed for regional-scale on Matlab@2016, utilizing both empirical studies and physical modeling for gravitational hazards (Horton et al., 2013). It is now widely applied and has achieved good results in different countries, for example, Michoud et al. (2012) simulated the road collapse disaster in the Swiss Alps; Blahut (2010) simulated the area affected by debris flows in Tirano, Italy. Rockfall modeling in Flow-R shows similar or even more realistic results than the other methods (Jaboyedoff et al., 2003). Michoud, Derron et al. (2012) reported that Flow-R software provided helpful results of rock block propagations for hazard mapping and risk assessment at a regional scale in the Swiss Alps. Losasso, Derron et al. (2016) used Flow-R to evaluate rockfall propagation extent and run-out distance in the Basilicata region, southern Italy. Flow-R can also be used to simulate other natural disasters, such as avalanches, debris flows, and floods (Horton et al., 2013).

In trajectory path simulation, the minimum reach angle or shadow angle is a key parameter controlling the fragment distribution results. Reach angle is suggested by Shreve (1968), Scheidegger (1973), and Hsu (1975), the mobility index H/L, where H is the fall height and L is the horizontal length of the landslide. And shadow angle is the dipping of the energy line which connects the farthest fallen boulder to the apex of the talus slope (Lied, 1977; Evans and Hungr., 1993). Many researchers have used it to study the propagation of rockfalls (Losasso et al., 2017; Kanari et al., 2019; Marchelli et al., 2019; Mitchell et al., 2020). Kanari et al., (2019) assessed the collapse risk by making statistics on the relationship between reach angle and rock size. Marchelli et al.(2019) used the relationship between rockfall size and slope to analyze the collapsing movement and rockfall fragmentation. In Flow-R, shadow angle is one of the key parameters in propagation assessment (Crosta et al., 2015). However, this angle and related modeling are not always calibrated probably due to data unavailability or budget constraints. The research works on rockfall have been mainly focusing on a static view of rockfall hazard or risk levels, without giving sufficient consideration to temporal probability for further quantitative risk assessment.

The mountainous areas, in general, face the problem of data scarcity mainly due to the inaccessibility of the terrain. Because of the scarcity of data and model uncertainty, we consider that the rockfall assessment methods including source area identification


and rockfall propagation should be at the level of probability assessment. According to rockfall terminology, rockfall hazard refers
to the probability of occurrence of an event (such as rockfall) of a given magnitude (such as volume) over a period of time within a
given area (Varnes et al., 1984; Fell et al., 1994; Guzzetti et al., 1999). The main objective of this research is rockfall hazard
probability assessment along the G318 national highway in China. Based on field investigation and satellite image interpretation,
historical rockfall hazards were inventoried and analyzed for slope threshold determination. Considering possible rock source
magnitude and rockfall event return period, hazard probability was simulated.
**2. Study area**

The research section of the national highway G318 is located in west Hubei, about 310 km Northeast of Chongqing, China

(Figure.1). It covers about 21.19 km$^2$. Intence erosion and weathering process created a cliffy topography in the southern part with
elevation ranging from 600m to 2000 m above sea level. Geological units in the study area (Figure. 1b, 1d) are mainly developed in
the Middle Jurassic stratum, except Triassic limestone partly covering at the anticline. Due to the wide syncline, the stratum is
mainly horizontal or gently deep in the area.

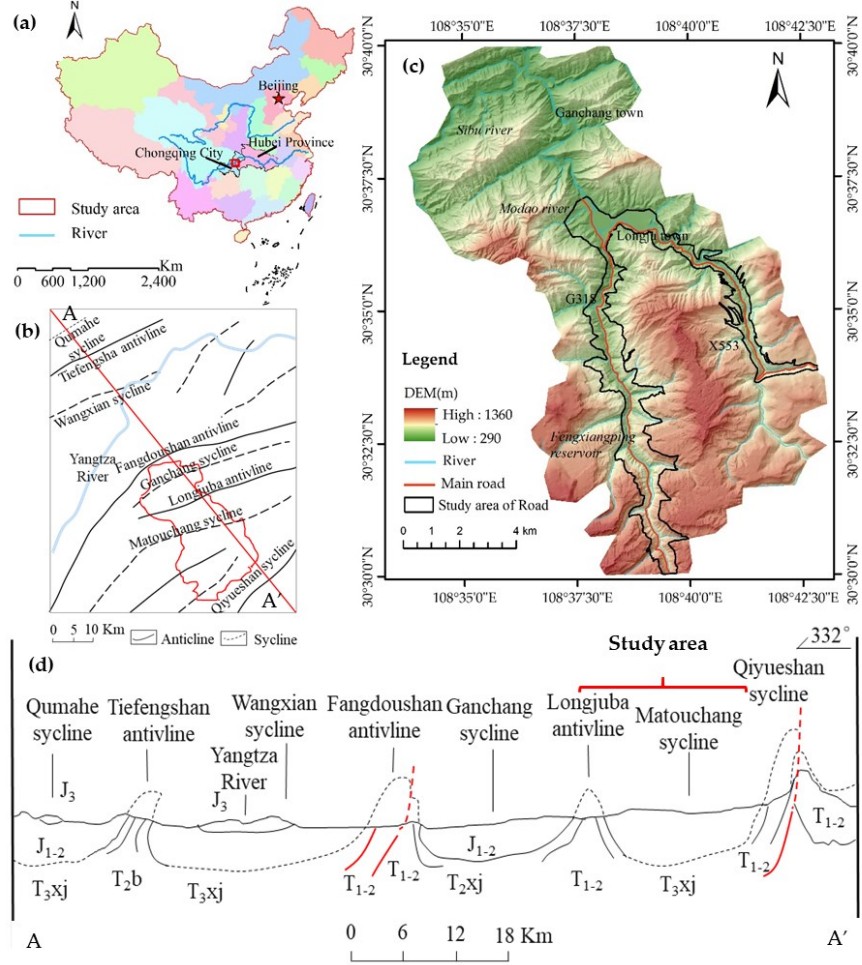


**Figure. 1.** (a) Location of the study area in the Qiyaoshan mountain ranges, in the middle part of China, (b) map showing the
geological structure of the study area; (c) Hillshade map of the study area; (d) geological structure profile along the cross-section
AA'.
Lithology in the area is purplish-red mudstone mainly, with sandstone and shell stone as interlayer, which has been affected
by physical weathering so that most rockfalls have taken place at these sections. National (G318) and provincial (X553) roads are
the main traffic ways, with a shape as an inverse Y across the area. Rockfalls occur frequently in the rainy season causing damage
to frastures as well as human casualties.
The north of Longju town is located in the Anticline of Fangdoushan and Jianchang syncline. The central part is the anticline
of Longju town and the syncline of Matouchang. Jiannan anticline and Jianzhuxi syncline are located in the south, so the tectonic
development of the study area is obvious. The rock strata in the core part of jianchang syncline are compressed and lithology is
dense. The Matouchang syncline is narrow and steep in the northwest and broad and gentle in the southeast, so it is near the
horizontal strata in the study area.
Rockfall is the main type of geological hazard in the area, especially in the Jurassic red bed (Middle Jurassic lithology) at the
nucleus or near-wings of the Matouchang syncline. Two sets of discontinuities control the rock quality and stability, combining
with the stratum layer face. Due to these controlling rock structures, differential weathering in sandstone and silty stone increases
the probability of rockfalls.
In the recent 10 years, urbanization of the Longjuba area in the Three Gorges dam area has been promoted by the government.
Accordingly, various construction works and reconstruction of transportation facilities have increased. In addition, due to the
construction of a new highway in the area, which involved cutting and filling the slopes, the Longjuba area is becoming more and
more hazardous, especially along the G318 (Figure.2. a), it can be seen that the highway collapse causes vehicle damage (Figure.2.
b), collapse dangerous rock mass (Figure.2. c) and small-scale collapse (Figure.2. d). According to the records of collapse disaster
database in the study area, the historical time of collapse in the study area is from 1984 to 2015.

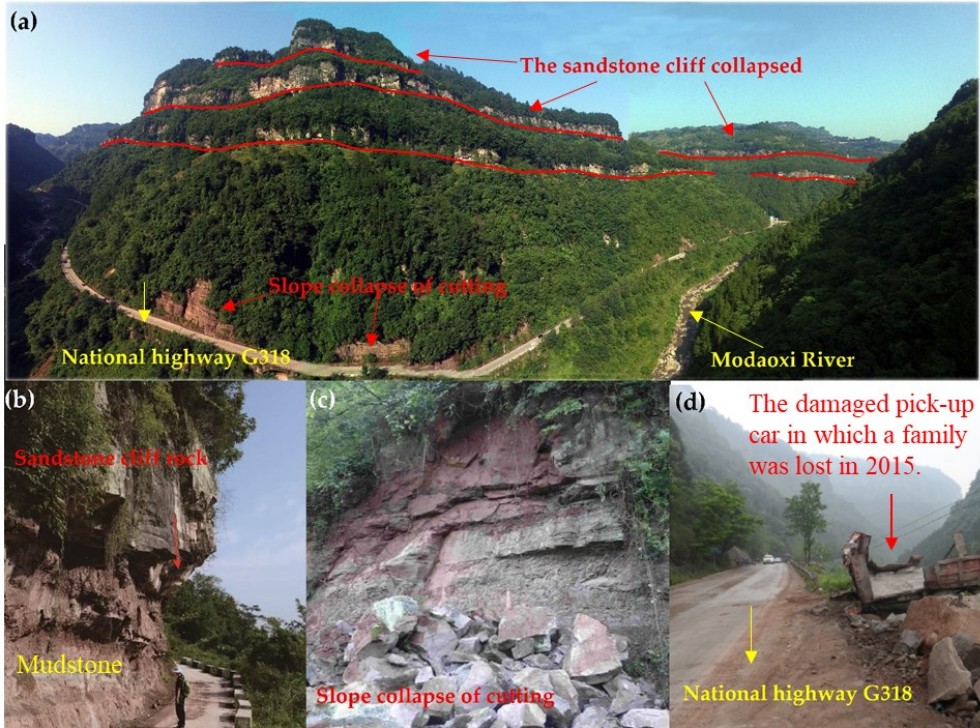

**Figure. 2.** Geological setting and natural hazards in the study area: (a) The cliff inter-bedding of sandstone and mudstone along
national highway G318 (UAVs image acquired in July 2016); (b) Sandstone cliff overlying mudstone along national highway
G318; (c) Falling down of the rock sources on national highway G318; (d) Damaged pick-up car and rockfall fragmentation.




**3. Methodology**

Rockfall hazard probability assessment was carried out following the flow chart shown in Figure. 3. Firstly, the study area was screened to extract the source area of collapse by using slope and topographic factors. On the basis of slope factors' analysis, Slope Angle Threshold (SAT) analysis is used to reduce the screening range of rockfall source areas. The most important inducing factors for collapse development were extracted from the obtained basic data to identify the source area of collapse. Multivariate logistic regression model (MLRM) and the random forest model (RFM) models were used and their results were compared by using the value of Area Under ROC Curve (AUC) to predict and identify the collapse source area. The spatial probability for the source area determination is simulated using Flow-R. The temporal and size probabilities were assessed using a historical rockfall distribution pattern. The methodology is described in detail as follows:

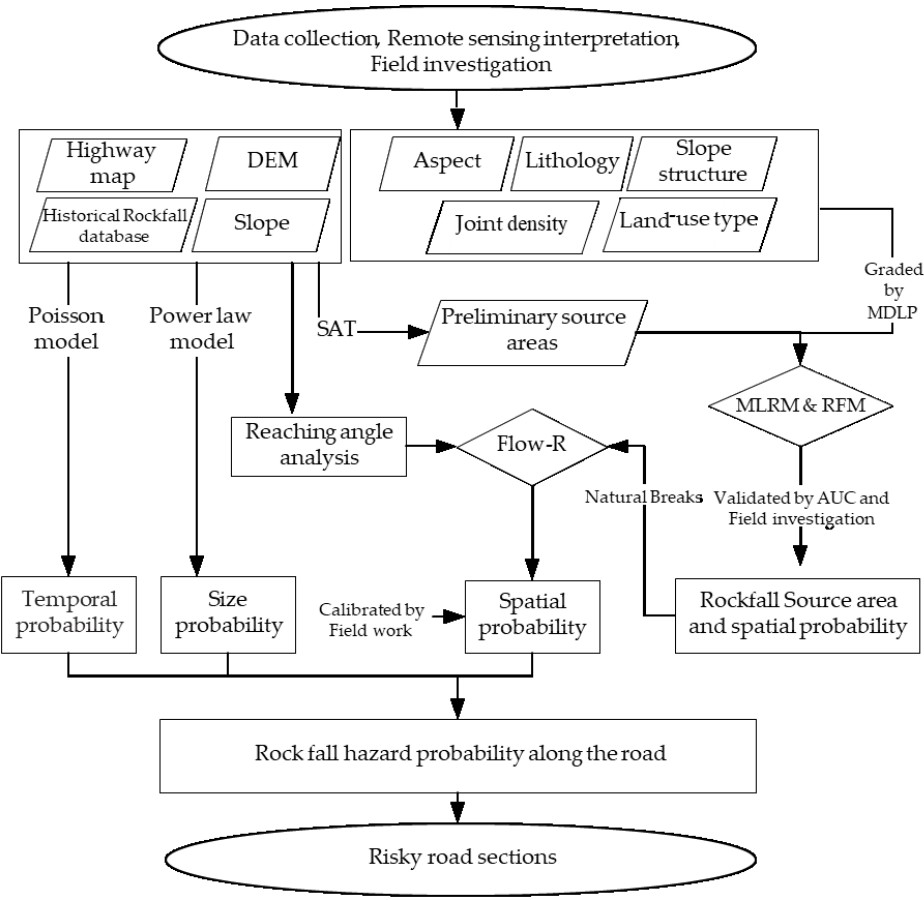

**Figure. 3.** Workflow of rockfall hazard probability assessment and risky road section identification

*3.1 Data collection*

The historical rockfall inventory map was generated by data collection (data from Wuhan Geological Survey Center, China Geological Survey Bureau), three years field investigation (2014-2016), and remote sensing image (Gaofen-1 data) interpretation. Among them, 20 were interpreted by remote sensing. In total 108 rockfall locations were identified, covering 31 years from 1984 to 2015. Among them, 31 rockfalls have precise volume data, which was later used for source area identification, temporal probability, and size probability analysis. Especially, the transportation characteristics (e.g. run out) are available for 37 rockfalls,


which were used for calibrating rockfall reaching probability simulation. There is no record of repeated disasters at all historical
collapse sites.
Besides the rockfall inventory data, other datasets were collected as follows:
● A 10m resolution DEM was generated from GaoFen-1 remote sensing data (resolution: 1m, image time: 2015.03.30), from
which slope, elevation, aspect, roughness, curvature, and solar radiation were generated using ArcGIS.
● A Geological map (1:10 000) was used to extract geological spatial layers such as lithology, faults, and slope structure map.
The slope structure map was generated using the standard and stratigraphic altitude advocated by Cruden (1991).
● The joint density data was gathered in the field in 2015. Joint sets were measured at 108 rockfall source areas.
● The land-use map was generated from the GaoFen-1 remote sensing data by applying the Spectral Angle Mapper
Classification method in ENVI software.
Specific data used are shown in Table 1.
**Table 1.** Source of data

| data | sources | resolution |
| --- | --- | --- |
| DEM | Gaofen-1 data (1m, image acquisition: 2015.03.30) | 10 m |
| Geological structure | Geological map | 1:50 000 |
| Lithology | Geological map | 1:50 000 |
| Joint density data | Field survey | 1:10 000 |
| Rockfall inventory data | Historical Data Collection, Field survey data and Gaofen-1 data | / |
| Remote sensing image | Gaofen-1 data | 1 m |

*3.2 Spatial probability of rockfall sources*
Rockfall sources are preconditions of rockfall hazards and risks. We need to determine the potential rocky slopes which have
the possibility to be unstable. In this study, three steps are recommended.
**Calculate the preliminary rockfall source area**
Firstly, we need to select the preliminary rockfall areas. In order to make a fine quantitative analysis of the collapse source
area, we need to digitize and resample the study area. According to the scope of the study area and the scale of the collapse, the size
of the grid is determined comprehensively. The preliminary source identification area of the collapse is constrained by the slope
angle threshold (SAT) method (Loye et al., 2009) in sequence. The SAT method can separate and remove the rockfall traveling area
and accumulation area. SAT method determines the slope threshold based on the relationship between the number of historical
collapses and the slope. The units with slopes steeper than the threshold are identified as preliminary rockfall source areas.
Secondly, rockfall conditioning factors in preliminary source areas are extracted and processed. The formation of collapse is
controlled by topography, physical and chemical weathering, human engineering disturbance, and other factors. Therefore, we
selected some factors that have the most serious impact on rock collapse in the study area. In addition to slope degree, the
determination of fine compounds source area is also constrained by slope aspect, elevation, lithology, slope structure (spatial position
between formation occurrence and slope face), joint density, land-use type, etc. Among these factors, slope, aspect, elevation, joint
density, and distance to roads are continuity factors. We use the minimal description length principle (MDLP) to classify these
continuity factors to improve the model prediction ability. MDLP is a method of discretizing continuous attributes, which has less
manual intervention and better quantitative effect (Varnes et al., 1984) than the methods such as equal frequency, equal width, and
artificial definition.
**Calculate the initial rockfall source area**
Then, the susceptibility of the preliminary rockfall source area is assessed and compared by the Multivariate Logistic regression
model (MLRM) and random forest model (RFM). In MLRM, the dependent variable is a dichotomic variable, with an absence-
presence value of a certain characteristic. In this study, this variable is historical rockfalls. The RFM is a mining method based on



statistical learning theory. It uses the idea of bagging to select a number of training samples and the establishment of a decision tree.
The output category is obtained by various categories of the voting output tree. The main advantages of RFM are random sampling
and features, avoiding overfitting, and improving the accuracy and stability of the model. RFM has achieved good results in the
field of early warning of geological disasters (Chen et al., 2014; Provost et al., 2017). To reflect the importance of each variable,
the Mean Decrease Gini (MDG) index was used. The higher the MDG index is, the more important the predictor (Liaw et al., 2012).
Before model prediction, rockfall source area and non-rockfall source area samples are prepared. Rockfall source areas are identified
as historical hazards. Non-rockfall source areas are randomly selected at least 500 meters away from rockfall source areas. We use
70% data of each group to generate a training dataset for model building and the remaining 30% for model testing. Using these
samples and the conditioning factors, rockfall susceptibility is modeled by MLRM and RFM. The performance of the two models
was evaluated.
**Obtain the final rockfall source area**
Finally, we classify the susceptibility value into five levels (very low, low, moderate, high, and very high) by the Natural
Breaks method. In this method, breaks are classified as large as possible between groups and as small as possible within groups.
The units with the highest class on the susceptibility map by the model with better performance are further finalized as rock fall
source areas.
*3.3 Temporal probability of rockfall sources*
The temporal probability of rockfalls is evaluated by assuming that rockfalls are independent random events in the time domain
(Crovelli et al., 2000; Fu et al., 2019). In this study, the Poisson model is adopted for constructing temporal probability. It is the
exceedance probability of rockfall occurrence during a given period as follows:

$$P_t = 1 - e^{-t/RI}, RI = T / N \tag{1}$$

Where $t$ is the return period, e.g., 5, 20, and 50 years; the recurrence interval ($RI$) is the historical mean recurrence interval for
each rockfall source unit; $T$ is the temporal interval of the rockfall database; $N$ is the number of historical rockfalls recorded in each
unit. Considering the possibility of missing rockfall points in the database, the units without historical records but having the highest
class of spatial probability in the source area susceptibility map are set as historical rockfall units.
*3.4 Size probability of rockfall sources*
Rockfall size probability is calculated by analyzing the relationship between rockfall volume and cumulative frequency. Bakp
et al. (1988) proposed that there is a certain power index relationship between rockfall volume and its frequency, which has been
verified in many regions (Pelletier et al., 1997; Malamud et al., 2004). This study follows the formula proposed by Malamud (2004)
to fit the size probability.

$$P(V; \rho, a, s) = \frac{1}{a\Gamma(\rho)} \left[ \frac{a}{V-s} \right]^{\rho+1} * \exp\left( -\frac{a}{V-s} \right) \tag{2}$$

Where $P_v$ is size probability; $V$ is rockfall volume; $\rho$ is parameter primarily controlling power-law decay for medium and
large values in three-parameter inverse-gamma probability distribution; $a$ is parameter primarily controlling location of maximum
probability in three-parameter inverse-gamma probability distribution; $s$ is parameter primarily controlling exponential rollover for
small values in three-parameter inverse-gamma probability distribution; $\Gamma(\rho)$ is the gamma function of $\rho$.
*3.5 Reaching probability of rock fragments to roads*
The probability of rock fragments from rocky sources is simulated by using the Flow-R software (Horton et al., 2013), which
can assess rockfall hazards with probabilistic trajectory paths at a regional scale. Rockfall source areas introduced in Section 3.2 are
input data in Flow-R. Besides, the rockfall trajectory path is determined by important input data, the reach angle (Wieczorek et al.,
1998; Guzzetti et al., 2003; Jaboyedoff and Labiouse, 2003; Frattini et al., 2008; Matasci et al., 2015). The reach angle is also known



as travel angle or travel distance angle (Copons et al., 2009; Duszyński et al., 2015), controlling rock fragments' probability and energy along the pathway. It is the arctangent of the line which connects the rockfall source area with the most distant boulder (Figure.4, Eq.3). The simulation assumes that the falling blocks stop at the point of intersection of the above-mentioned line with the topography where the energy is 0 (Copons et al., 2009).

$$\theta = \arctan\ (H/L) \tag{3}$$

Where $\theta$ is the reach angle, which is from the vertical drop $H$ and the horizontal component of the travel distance $L$. The longer the travel distance is, the lower the reach angle value will be.

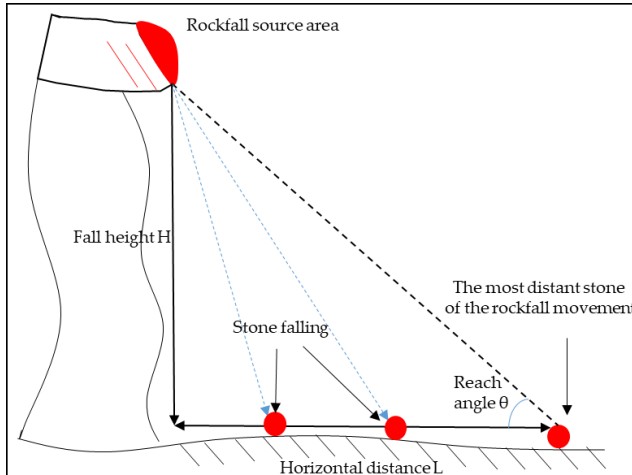

**Figure. 4.** Reach angle diagram

*3.6 Rockfall hazard probability assessment*

The purpose of rockfall hazard assessment in this study is to know the possibility of rockfall fragments reaching the road with a certain magnitude under a certain return period. We multiply four probabilities to assess the hazard level (Eq.4). By overlaying the hazard probability map with the highway map, risky road sections can be identified finally.

$$H = P_s \times P_t \times P_v \times P_r \tag{4}$$

Where $H$ is rockfall hazard probability; $P_s$ is the spatial probability of rockfall sources introduced in Section 3.3.1, $P_t$ is the temporal probability of rockfall sources, $P_v$ is size probability of rockfall sources; $P_r$ is reaching probability of rockfall sources to roads.

*3.7 Validation*

The generated susceptibility maps by MLRM and RFM were validated by using the Receiver Operating Characteristics (ROC) curve (Cruden et al., 1991; Zezere et al., 2017) and expert re-evaluation on for typical slopes in the field. The larger the value of area under the ROC curve (AUC) is, the more effective the evaluation result is. The performance of the two models was measured and compared. The units with the highest class susceptibility by the better model will be determined as rockfall source areas. The temporal probability of rockfall sources validation is not easy to be taken due to limited data on hazard occurrence time. Whereas the size probability of rockfall sources is validated by calculating the R-squared value of the exceeding probability distribution curve.

Parameters for rockfall reaching probabilities are firstly calibrated and determined by repeating trials in Flow-R on two historical rockfall events with detailed run-out measurements. Then the selected parameters are further validated by simulating the





other 35 of the 108 historical rockfalls with accumulation area information. Pictures of historical rockfalls by UVA are also used to
verify the accuracy of runout distance and reach angle.

**4. Results**

*4.1 Rockfall source area determination*

Using the SAT method, the slope angle threshold for the study area is determined as 27° (Figure.5). This value is smaller than
the research result by other researchers (e.g. Wieczorek et al., 1998, and Guzzetti, Reichenbach, et al., 2003; Jaboyedoff and
Labiouse; Frattini et al., 2008; Matasci, Jaboyedoff, et al., 2015). It underlines that the areas with different topography or geology
characteristics do not have the same SAT value. Based on this SAT value, 763 847 cells with a slope greater than 27 ° are selected
as preliminary rockfall sources from the total area with 1 443 012 cells.
Figure.5 shows the relationship between slope factor and collapse disaster with 3 ° step size. In Figure.5, the rockfall area ratio
equals the source area of rockfall within a certain slope degree range divided by the total rockfall areas; graded area ratio equals to
the slope within certain slope degree divided by the total study area. When the rockfall area ratio is in the interval of [0°, 27°), the
graded area ratio begins to be greater than the rockfall area ratio. About 80% of rockfalls are concentrated in this section. After 27°,
the graded area ratio begins to be greater than the rockfall area ratio. Therefore, the area with a slope greater than 27 ° is selected as
the preliminary rockfall source.

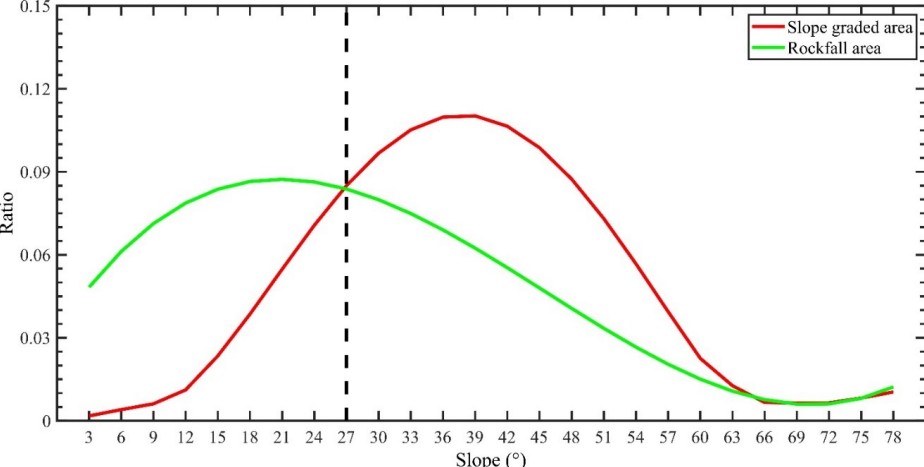

**Figure. 5.** Relationship between rockfall distribution frequency and slope

The preliminary rockfall sources are further classified by considering eight conditioning factors, such as slope, aspect, elevation,
roughness, slope structure, lithology, etc. Table 2 lists out these factors with classes by using MDLP (Varnes et al., 1984).





**Table 2.** Rockfall source area conditioning factors with classes by using minimal description length principle

| Conditioning factors | Classes | Conditioning factors | Classes |
|---|---|---|---|
| Slope (°) | | Slope structure | Over-dip slope |
| | 27 - 34 | | Under-dip slope |
| | | | Oblique slope |
| | 34 - 38 | | Tranverse slope |
| | 38 - 47 | | Anaclinal slope |
| | | Lithology | Soft rock or soft and hard interbedded rock |
| | 47 - 51 | | Soft rock with hard rock |
| | 51 - 79 | | Hard rock with weak interbed |
| Aspect (°) | North | Joint density (A/m) | 0 - 0.68 |
| | Flat-North-Northeast | | 0.68 - 0.8 |
| | Northeast-East | | 0.8 - 1 |
| | East-Southeast | | 1 - 1.5 |
| | Southeast-Southwest | | 1.5 - 3 |
| | Southwest-Northwest | | 0 - 32 |
| | Northwest-North | Distance to road (m) | 32 - 71 |
| Elevation (m) | 290 - 368 | | 71 - 247 |
| | 368 - 426 | | 247 - 260 |
| | 426 - 474 | | 260 - 364 |
| | 474 - 549 | | 364 - 410 |
| | 549 - 674 | | ≥410 |
| | 674 - 1101 | Land-use type | Grasslands and Open Wood |
| | 1101 - 1139 | | Rock and Exposed Soil |
| | 1139 - 1360 | | Water |
| | | | Rural Settlement |


Figure.6 shows susceptibility maps of rockfall source area by MLRM and RFM. In terms of the ranking of importance of the
factors, distance from road and slope is the most important as shown in both the models (Figure.7). However, a big difference exists
in lithology and land use. RFM ranks lithology as a relatively insignificant predictor but this factor is treated to be the third important
in MLRM. As to the land-use factor, it is not effective or the least significant in the ranking in both models.
ROC curve analysis shows that the success rate of MLRM is 93%, while RFM is 5% higher (Figure.8). It indicates that RFM
has a better model performance than MLRM in the study area. The prediction performance of the two models was further evaluated
and compared in the field (Table 3).
Four typical slopes along G318 road were selected for validation, including steep slope with sandstone inter-bedding with
mudstone, gentle rocky slope, steep slope with high vegetation cover, and gentle rocky slope with high vegetation cover. Each slope





was evaluated by the experts resulting in the possibility of rockfall source potential. In the comparison results of the susceptibility
of four typical slopes, the RFM results of the susceptibility of three slopes were consistent with the expert judgments. It further
shows that RFM has a better evaluation effect.
Using the result from RFM, rockfall source area spatial probabilities are finally divided into four classes: [0, 0.25), [0.25, 0.60),
[0.60, 0.88) and [0.88, 1]. The percentage of historical rockfalls in each class is shown in Table 4. A region with a spatial probability
of [0.88, 1] from RFM was finally selected to simulate rock fragment trajectories. This allowed identifying 5349 grid cells (10×10
m) as final sources of rockfalls, about 0.53 km$^2$ (0.70% of the total study area). Inspection of the map of the rockfall source cells
revealed a good agreement with the local morphology, and in particular with the location of the edges of the rock cliffs and with the
location of the release areas of known rockfall events.

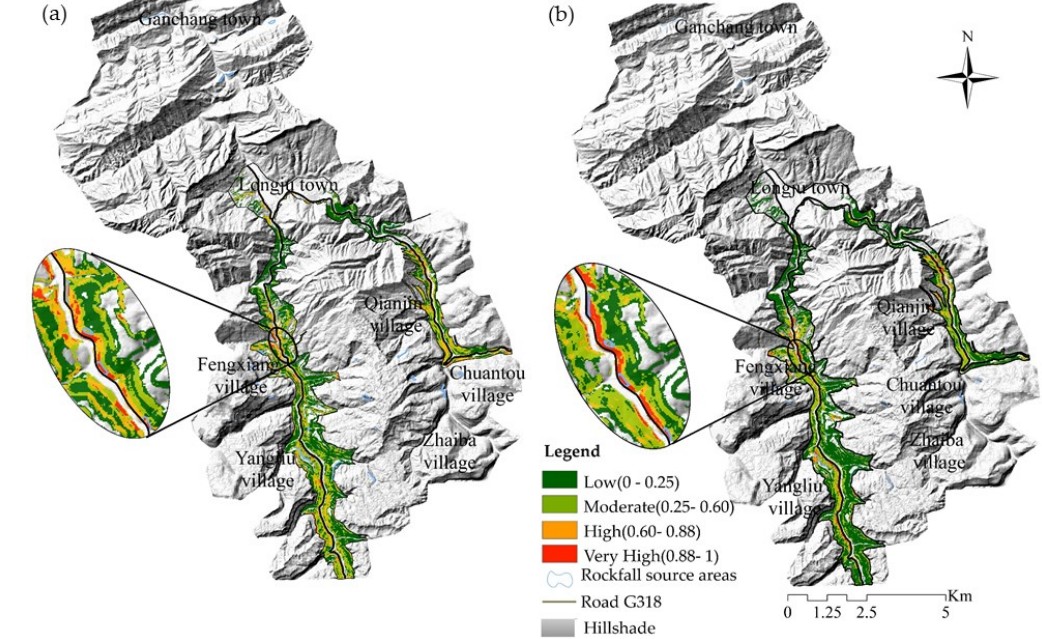


**Figure. 6.** Spatial probability maps of rockfall sources by (a) MLRM and (b) RFM

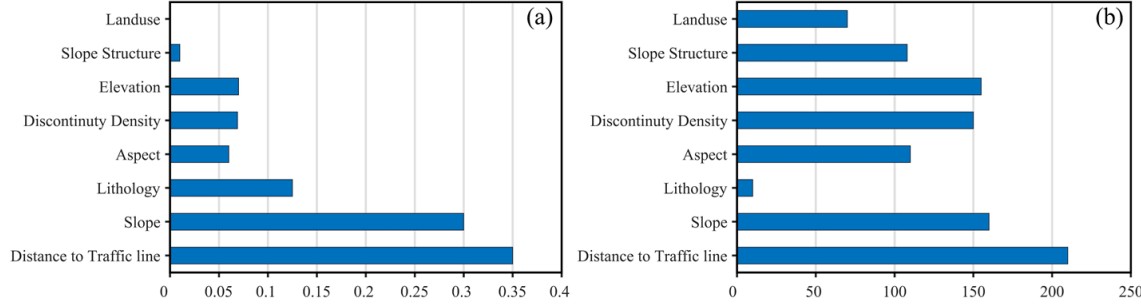


**Figure. 7**. Ranking of predictor factors by (a) MLRM and (b) RFM.




Figure. 8. Accuracy comparison between MLRM and RFM


**Table 3.** Comparison of rockfall source area probability result from MLRM, RFM, and field evaluation

(The legend is the same as in Figure.6)

| Number | MLRM | RFM | Field photos | Expert evaluation |
|---|---|---|---|---|
| No.28 | | | | The integrity of the rock mass is good, but there are blocks piled up on the slope. It is a high-medium class. The result from MLRM is accurate. |
| No.30 | | | | The slope surface is gentle, and there is no rockfall accumulation, which is a middle-class-prone area. The result from RFM is accurate. |
| No.31 | | | | The vegetation coverage rate is high and the slope surface is gentle. It is a low-class prone area. The result from RFM is accurate. |
| No.32 | | | | The vegetation coverage rate is very high. There are no exposed rock blocks. It is a medium-low class prone area. The result from RFM is accurate. |








**Table 4.** The Proportion of historical source area in each probability grades classified by RFM (Area: km$^2$)

| Probability grade | A- Area(percentage) | B- Historical rockfall source area (percentage) | B/A- Proportion of historical rockfall source areas in different probability grades (%) |
|---|---|---|---|
| 0 -0.250 | 68.140 (89.220%) | 0.060 (28.070%) | 0.088 |
| 0.250-0.600 | 6.400 (8.390%) | 0.040(17.390%) | 0.625 |
| 0.600-0.880 | 1.290 (1.700%) | 0.040 (18.530%) | 3.101 |
| 0.880-1 | 0.535(0.700%) | 0.080 (36.010%) | 15.094 |
| Sum | 76.365(100%) | 0.220(100%) | 18.908 |


Quantitatively, the simulation efficiency of our approach can be improved by 40 times, without losing data of historical or
field survey determined rockfalls. Detailed data are shown in Table 5.
**Table 5.** Comparison between SAT model and our approach in terms of simulation efficiency

| | SAT model | Our approach | Benefit or Loss |
|---|---|---|---|
| Number of potential source cells | 160823 | 4002 | Benefit: about 40 times |
| Number of historical or field survey determined rockfalls | 1364 | 1345 | Loss: about 0.014 times |


*4.2 Temporal and size probability of rockfall sources*
According to the records of the rockfall database in the study area, the historical time is 31 years. According to Equation 1,
temporal probabilities in different recurrence periods (5, 20, 50 years) for each unit are calculated and the relative maps were
generated (Figure.9). The map for 50 years, for example, shows that the highest temporal probability is 0.798 (Figure.9a). In order
to enhance the distinction, the temporal probability maps are divided into five classes by the Natural Breaks method. The areas with
the highest class with temporal probability from 0.536 to 0.798 mainly distribute along the road G318 in the north part.

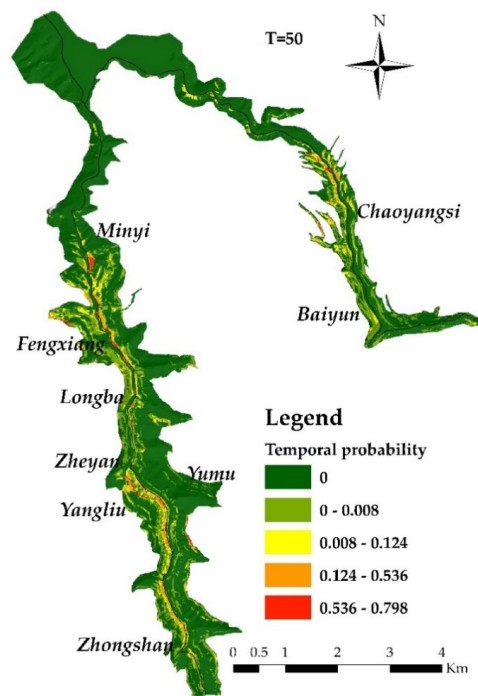


**Figure. 9**. Fifty years return period maps of rockfall temporal probability along the national road G318

By using 31 historical rockfalls with volume records, the size probability curve is created according to Equation 2 with an $R^2$ value of 0.956 (Figure.10). The rockfall volume ranges from 1 000 $m^3$ and 10 000 $m^3$ with size probability from 0.826 and 0.395, which means: (1) the occurrence probability of rockfalls with volume greater than 1 000 $m^3$ in the study area is 0.830; (2) the occurrence probability of rockfalls with volume more than 10 000 $m^3$ is 0.395. It indicates that small-scale rockfalls are more frequent than the larger ones, which is consistent with the real performance of rockfall hazards in the study area.

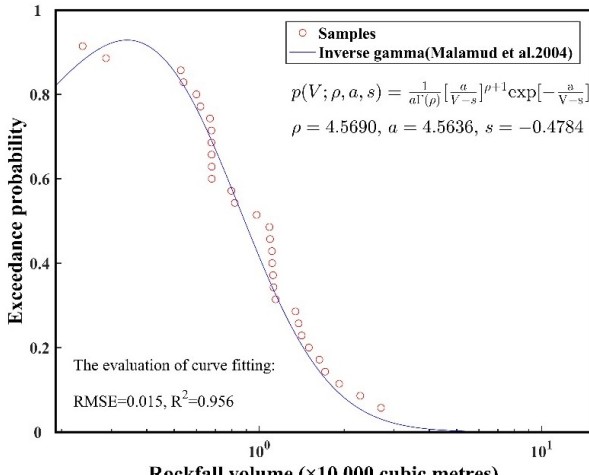

**Figure. 10.** Size probability distribution curve fitted by inverse gamma for rockfalls along the national road G318




312   According to Equation 4, the hazard probability maps are generated for three return periods (5, 20, and 50 years) and two-

313 volume scenarios (10 00 m³ and 10 000 m³). The probability values of 10 00 m³ volume scenarios in Figure.9b comprise five

314 categories from very low (0 - 0.112) to very high (0.538- 0.801). The area with obvious high probability is close to the road section

315 from Minyi to Fengxiang. It is located in the south of Longjuba anticline and the core part of Matouchang syncline, with strong

316 geological tectonic activities. Lithology in the area is mainly covered by purplish-red mudstone, sandstone, and shell stone as

317 interlayers. Rocks are seriously weathered in this section.

318 *4.3 Rockfall reaching probability and verification in field work*

319   Table 6 summarizes the algorithms and parameters used in Flow-R by repeating trials on two historical rockfall events

320 (Figure.11).In terms of reach angle, we found three possible values (15 °, 25 °, and 27 °) according to the law of reach angle

321 distribution of historical rockfalls. To find out the most suitable value, we compared the simulated travel distance with the measured

322 value in the field (Figure.11). Therefore, reach angle 25 ° is adopted as the preliminary reach angle value for further verification in

323 other 35 rockfall modelings. The further simulation results show that the average difference value is 1.97 m with an evaluation error

324 ratio of 3.66% (Figure.12). The simulated reaching area basically matches the influence areas where rock fragments are scattered

325 (Figure.13). It indicates that simulated travel distance fits well with the value investigated in the field by using a reaching angle of

326 25 °.

327      **Table 6.** Initial modeling conditions and parameters used in Flow-R to perform the transportation simulation

| Algorithms and Parameters | Value |
|---|---|
| Flow direction algorithm | Holmgren modified algorithm |
| Exponent α | 1 |
| Persistence factor | Gamma_ 2000 |
| Friction model | Simple Coulomb friction model |
| Minimum reach angle | 25 ° |

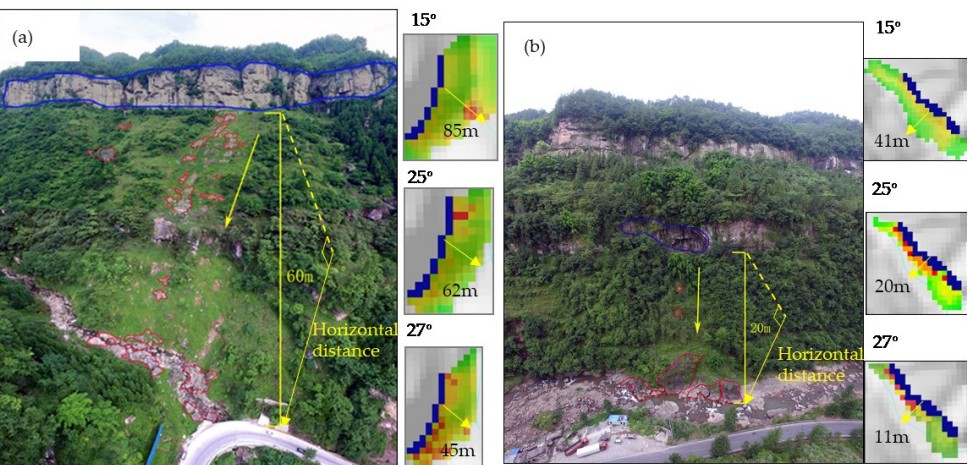

329 **Figure. 11.** Comparison of horizontal travel distance between the field measurement and numerical simulation for reach angle

330 determination. (The source area is pointed out using blue closed lines. Rockfall fragments are identified using red closed lines.

331 Horizontal travel distances marked by yellow lines.)


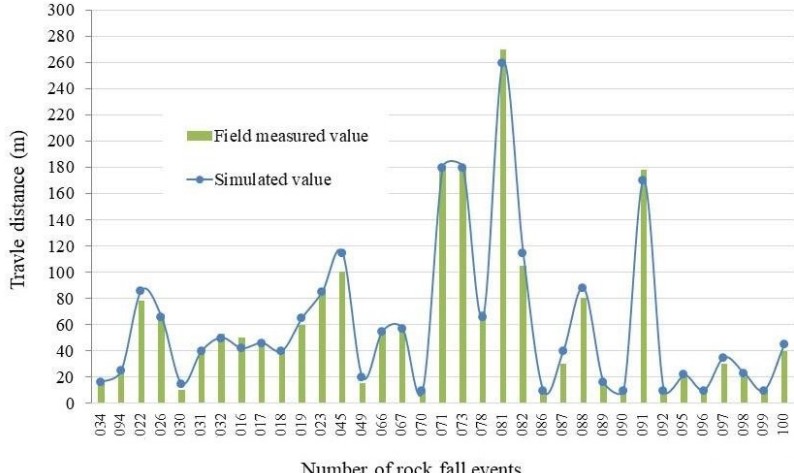

**Figure. 12.** Horizontal travel distance comparison between the field measured value (green column) and (b) the simulated value
(blue line) for 35 historical rockfalls along the national road G318.

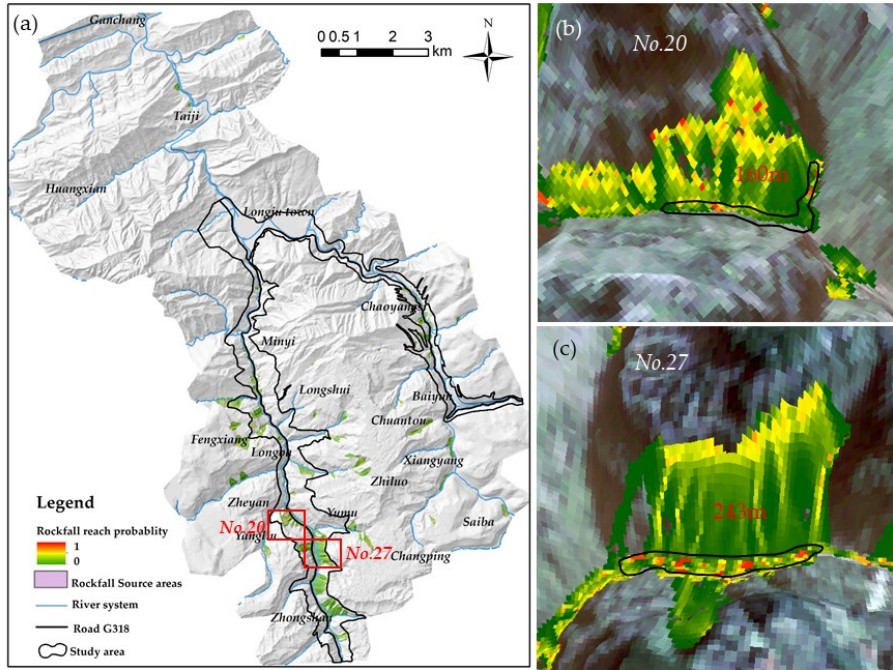

**Figure. 13**. (a) Reaching probability map of rockfall along the national road G318. (b)(c) Enlarged view of partial collapse
motion. (Rockfall fragment distribution area is identified using black closed lines)

*4.4 Rockfall hazard probability along the road G318*

Rockfall hazard probability was calculated according to Equation 4 by overlapping the maps of spatial (Figure.6), temporal
(Figure.9) and size probability of rockfall sources, and the reaching probability map (Figure.13). The final maps (Figure.14) of
rockfall hazard probability were set for two size scenarios and three return periods. When the volume scenario is 1 000 m³, the




maximum probability value increases from 0.123 to 0.661 with the increase of return period (Table 7). Because of the lower size
probability of 10 000 m³, the maximum hazard probabilities are generally half of the values under the size scenario 1 000 m³.
If the above results are associated with the national road G318, we can find out the risky sections with detailed impact
probability for road G318 due to rockfall fragments. Table 8 lists the length of the impacted roads under each return period and
volume scenario. Among them, Minyi village, Longba village, Zheyan village, Yumu village, and Zhongshan village are located
along the 318 national highway and 553 county roads in Chaoyangsi village and Xiangyang village are the most affected by the
rockfall, so the protection and control should be strengthened. G318 section with high hazard is mainly located in Minyi Village
and Zheyan Village.
In general, for different return periods and collapse scales, the total length of damaged sections is 8.19 km, and the damage
degree of collapsed roads in the 50-year return period is higher than that in the 20-year return period and the 5-year return period.
The severity of road damage caused by the collapse disaster with a scale of larger than 1000 m³ is higher than that of the collapse
disaster with a scale of larger than 1000 m³. In the return period of 5 and 20 years, there is no very-high hazard class of road section,
but mainly concentrated in the rainfall conditions of 50 years return period, and the influenced road section caused by the collapse
of larger than 1000 m³ is 0.510 km, and the influenced road section caused by the collapse of larger than 10,000 m³ is 0.430 km. In
the 5 years of the rainfall return period, there is no high-class risk of road section but mainly concentrated in the rainfall conditions
of the 50 and 20 years of the return period.

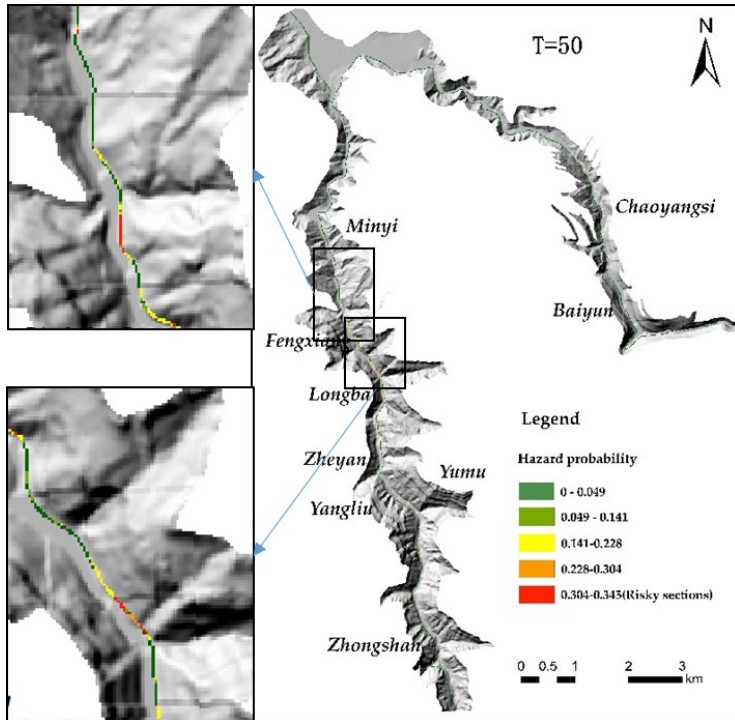


**Figure. 14**. Rockfall hazard probability map (at the volume scenario of 10 000 m³) for the national road G318. The map is
classified into five categories from high to low, which are overlapped by the road sections.





**Table 7.** Maximum hazard probability under three return periods (5, 20, and 50 years) and two-volume scenarios (1 000 m³ and 10 000 m³)

| Volume scenario (m³) | Return period (years) | | |
|---|---|---|---|
| | 5 | 20 | 50 |
| 1 000 | 0.123 | 0.393 | 0.661 |
| 10 000 | 0.059 | 0.188 | 0.287 |

**Table 8**. Influence length of road G318 induced by rockfalls (unit: Km) with two size scenarios and three return periods

| Size scenario / Return period (years) | Very High | | High | | Medium | | Low | | Very low | |
|---|---|---|---|---|---|---|---|---|---|---|
| | 1 000 | 10 000 | 1 000 | 10 000 | 1 000 | 10 000 | 1 000 | 10 000 | 1 000 | 10 000 |
| 5 | 0 | 0 | 0 | 0 | 0 | 0 | 0.510 | 0.480 | 7.680 | 7.710 |
| 20 | 0 | 0 | 0.430 | 0 | 0.090 | 0.510 | 1.490 | 1.500 | 6.180 | 6.180 |
| 50 | 0.510 | 0.430 | 0.010 | 0.080 | 1.480 | 0.010 | 0.010 | 1.490 | 6.180 | 6.180 |

## 5. Discussion

In understanding and analyzing rockfall hazard risk, it is very important to identify the source areas, predict the temporal, size, and reaching probability.

### 5.1 Difficulties in identifying source area of rockfall

The identification of the source area is the first step. The fineness of source area identification has an important impact on the following steps, such as the fragment trajectory and rockfall size analysis. However, the source area of historical rockfall hazard data is often missing or mixed with the rock debris accumulation, so it is difficult to identify the source area. Luckily, the slope angle threshold is found out to be 27°in this study, according to the relationship between the historic data and the slope. The area above this angle is preliminarily selected as source areas. After the preliminary screening of the collapse source area in the study area by using SAT method, we conducted a secondary screening of the initial source results in the study area by using various models. By using and comparing multivariate Logistic regression model and random forest model, the final source areas are determined and had a good accuracy after validation. Importantly, the efficiency of trajectory simulation followed by our approach can be improved by 40 times, without losing data of historical or field survey determined rockfalls.

Due to the special topography and geological conditions, there are a large number of multi-stage scarps in the study area (as shown in Figure. 15), and more accurate source area identification is required. In the future, more detailed work will be focused on the source stage scarp identification.

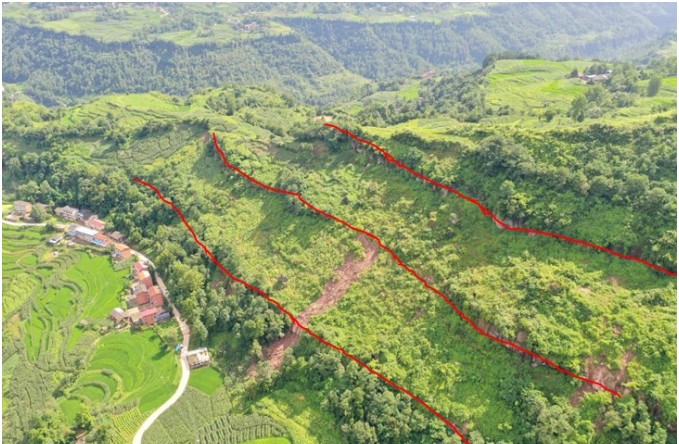

**Figure. 15** UVA photos of multistage rockfall
*5.2 Complexity and difficulty in the time probability calculation*
The temporal probability and the size probability are important considerations in rockfall hazard analysis. In practice, the
temporal probability calculation is a difficult problem, on the premise that there should be a large number of historical collapse time
data to analyze the statistical law. However, due to the sudden occurrence of collapse, it is difficult to obtain a large number of time
data, which requires a lot of monitoring work. The same is true of the probability of scale surpassing, which requires the scale data
of every rock collapse in history for statistical analysis. Rockfalls in this area mostly happen along the traffic road. For road
accessibility, rock fragments are quickly cleaned after hazard events so that historical influence area record is always unavailable.
Both of them have always been difficult points in collapse hazard analysis.
It is difficult to calculate the time probability for multistage cliffs. There are a large number of historical rockfalls in the study
area, such as the PT rockfall in Figure 16. The first occurrence time of PT rockfall is June 29, 2019. The second occurrence time is
July 5, 2020. This kind of multi-stage collapse disaster causes serious economic loss and great psychological pressure on the victims'
families. Therefore, it is necessary to solve the problem of how to accurately predict the time and calculate the time probability of
multiple collapse disasters of multistage cliffs. But we need to do long-term monitoring and collect large amounts of historical
occurrence time data for predicting these types of collapses. So, the establishment of a historical collapse time database in the study
area is needed in the future.

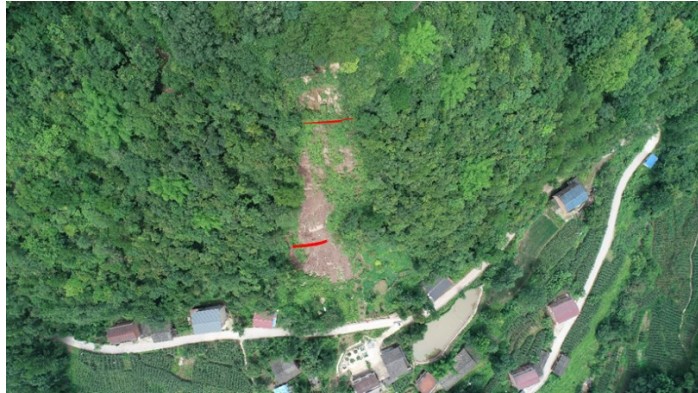

**Figure.16** A typical case of multiple collapses in the study area, PT collapse (The first occurrence time of PT collapse is June 29,
2019. The second occurrence time is July 5, 2020.)



*5.3 Simulation problem of hazard calculation*

This paper adopted energy balance theory, GIS spatial statistical function, and flow theory to simulate the influence area of rock fragments. The parameters in the simulation are calibrated and validated by historical records collected by field investigation. The results indicated that the accuracy of the quantitative analysis is very high. However, the failure motion of collapse is various, which was ignored in the Flow-R simulation. There are multiple failure modes of collapse, such as dumping, falling, and sliding. The simulation procedure simplifies the laws governing rock-mass failures and blocks propagations.

Compared with STONE, Rockyfor3D, RAMMS, DDA, Flow-R can simulate the motion of multiple collapsing sources on the regional scale by using less time and costs. But we can not consider the failure modes by Flow-R tools. In the future, we will optimize the simulation considering rock source volume, block shape, failure modes, and mechanical parameters and achieve a three-dimensional dynamic display of the collapse process at the regional scale.

The simulation of multistage scarps should consider the energy transfer caused by the collision between the scarps or the induced collapse of the scarps. For example, in Figure. 16 PT collapse is induced by the falling of a boulder in the upper layer. For the complexity of collapse, more research work is needed in the future.

## 6. Conclusion

A national road G318 in west Hubei China is prone to the high-frequency rockfall hazard. In this paper, rockfall hazard and its probability are quantitatively assessed. Rockfall source areas are firstly identified by the slope angle threshold method and then optimized by using the susceptibility mapping method. Slope degree 27° is determined as the threshold angle of rockfalls in the study area. The multivariate logistic regression model and random forest model are compared in terms of the model performance. Source area cells selected by the random forest model are finally chosen and applied for rockfall reaching probability assessment. Compared to the slope angle threshold method, the source areas determined by our approach are more accurate when geology data is available. Meanwhile, the advantages of trajectory simulation efficiency are obvious and without losing data of historical or field survey determined rockfalls. In addition, the size probability and temporal probability for rockfall sources are calculated considering two size scenarios (1 000 m3 and 10 000 m3) and three return periods (5, 20, and 50 years).

The selection of parameters is very important for the rockfall trajectory simulation. The smallest reach angle affects the farthest horizontal distance and then the reaching probability. In this paper, 25 ° is determined as the smallest reach angle. The horizontal distance is then simulated by Flow-R and then validated with the historical rockfalls with field-measured records. In the future, we will optimize the simulation considering rock source volume, block shape, failure modes, and mechanical parameters and achieve a three-dimensional dynamic display of the collapse process at the regional scale.

Rockfall hazard probability is finally obtained by integrating the spatial, temporal, size probability of source areas and the reaching probability of rock fragments. In the rainfall return period of 5 and 20 years, there is no high hazardous road section, but they are mainly concentrated in the conditions of 50 years return period. In this case, the risky road section caused by rockfalls larger than 1000 m³ is 0.510 km. Among them, villages including Minyi village, Longba village, Zheyan village, Yumu village, and Zhongshan village are identified along the national road G318, so the protection and control are suggested in these villages. Although some limitations exist, the results show good fitness with the measurements by field investigation.





**7. Patents**

**Author Contributions:** Conceptualization, Lixia Chen, Kunlong Yin; methodology, Lixia Chen, Yu Zhao, Lei Gui; investigation, Lixia Chen, Yuanyao Li, Lei Gui; writing, Lixia Chen, Yu Zhao; writing—review and editing, Dhruba Pikha Shrestha; visualization, Yu Zhao,Lixia Chen, Lei Gui; All authors have read and agreed to the published version of the manuscript."

**Funding:** This research was funded by the National Natural Science Foundation of China (No.41877525).

**Acknowledgments:** We would like to thank the editor and anonymous reviewers providing valuable contributions and constructive comments.

**Conflicts of Interest:** The authors declare no conflict of interest.

**Appendix A**

**Factors**

Eight factors including slope, aspect, elevation, slope structure, lithology, joint density, land-use type, and distance to the road are selected for source area identification (Figure.A1).

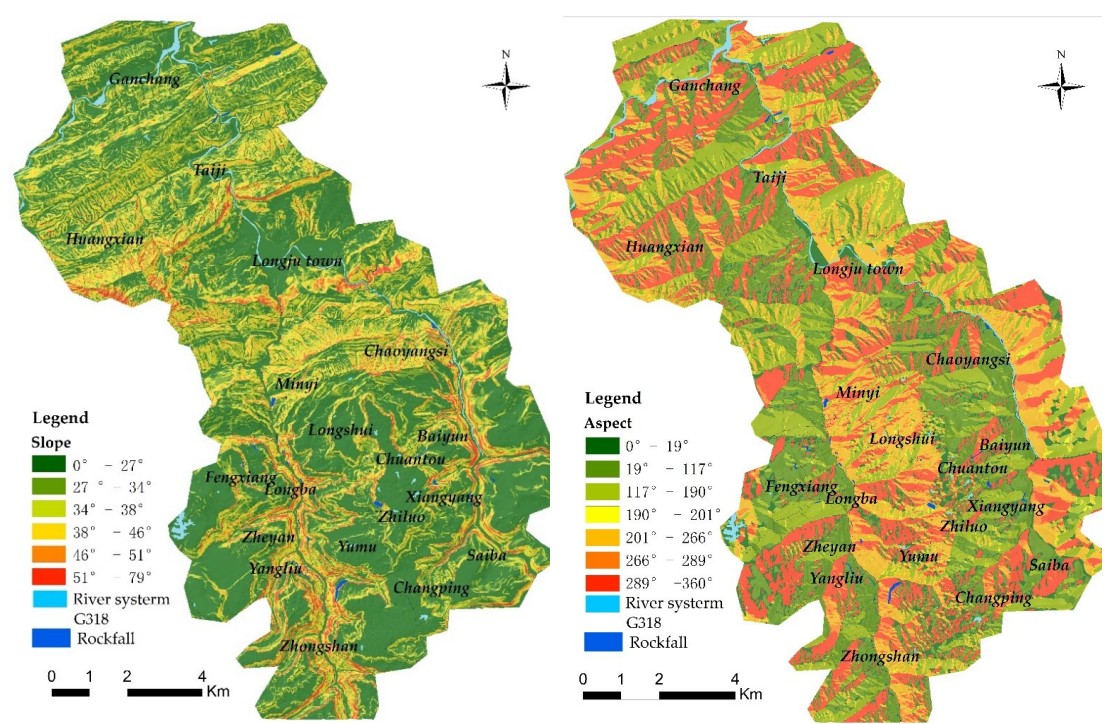





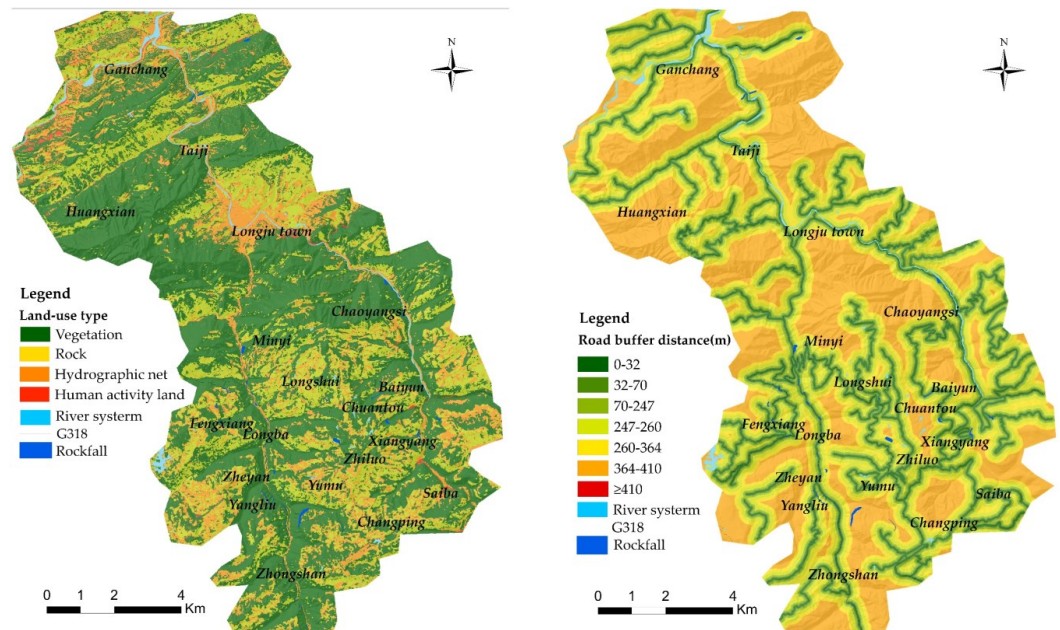

**Figure. A1.** Conditioning factor maps for rockfall source area identification

The complete result of the hazard probability is shown in Figure.A2. Figure.A2 includes the hazard probability results in the
10-year and 20-year return periods.

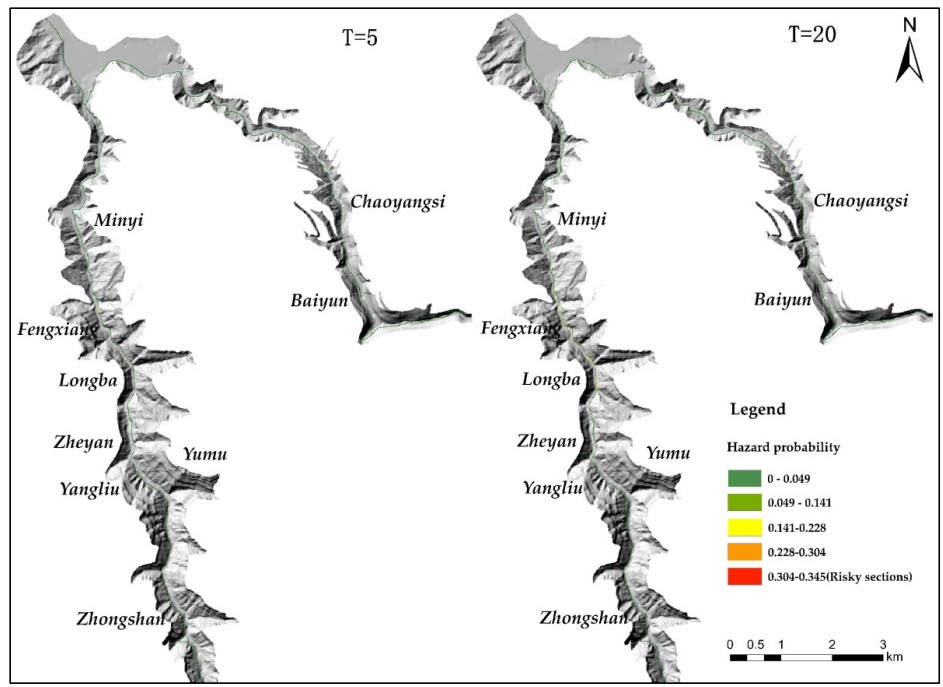

**Figure. A2.** Rockfall hazard probability assessment (at the volume scenario of 10 000 m$^3$) in the 10-year and 20-year return
periods for the areas along national road G318 in Longjuba.





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
