# Peer review of "Understanding rockfalls along the national road G318 in China: from source area identification to hazard probability simulation"

_Natural Hazards and Earth System Sciences, 2021_

## Referee Comment (RC1)

Comments to the authors of the manuscript under discussion entitled "*Understanding rockfalls along the national road G318 in China: from source area identification to hazard probability simulation*"

**Key finding**: The presented manuscript deals a comprehensive assessment for rockfall hazard along the national road G318 in west Hubei. It combines geological surveys, disposition analysis and trajectography with Flow-R as input for rockfall hazard probability calculations.

**Main concerns**: The paper is a mash-up of many techniques in rockfall hazard classification/risk assessments. It starts out with rather simple and straightforward slope angle threshold techniques (executed incorrectly), adds statistical models/ such as minimum description length (MDL), multivariate logistic regression model (MLRM) and the random forest model (RFM). The flawed SAT method unfortunately leaves the comparison with more elaborate models useless. Although MLRM and RFM models may serve as additional tools, the presentation in this publication is not stringent and the sole idea of mashing all methods together does not provide a better result.

Honouring Occam's razor, if the authors want to pursue all of the mentioned techniques, they need to verify for each model addition, why it is performing better than a simpler methodology. As they fail to perform the first task of the SAT threshold analysis in a correct way, the reasoning for more complex models are insubstantial.

The calculation of rockfall hazard probability where size scenarios are independent of return periods shows a fundamental misunderstanding in combining probabilistic methods.

Combined with the excessive incorrect placement of questionable, old and mis- referenced sources, the overall manuscript – although it is based on a solid field examination and an intrinsically valuable data set – fails to deliver scientific thoroughness and quality sought for in this journal.

The authors are urged to re-assess their tool sets, assess performed analysis with great care and fit their work in a thourough background. Name dropping and big data approaches with small data sets and inconsistent analysis and error checks does not help the scientific community (and is a waste of the reviewers valuable time).

After all, I cannot recommend this article for publication. A resubmission would require substantial re-thinking of the applied methods, thorough cross-comparison of the individual tools and in-depth validation for each complexity introduced.

In the following there are some minor/technical/content corrections: (aborted after too many inconsistencies in the conception/execution/writing made a recommendation for publication impossible)

**Technical corrections:**

General comment on the use of parentheses: Clearly a matter of writing style, however, IMHO the excessive use of parentheses hinders the reading flow. Personal guidance is: if it's important, rephrase it into the written sentences, if it does not merit being included in the text, remove it. The authors might check their use of parentheses with this in mind, or discard it as the referee's spleen. Does not hold for introduction of acronyms, of course.

Figure font sizes: Revise the font size and general sizing of heavily loaded figures.

**Abstract:**

l14: kinemics → kinematics, but is it really?
l20: results agree with measurements, fit well the acquired field data, etc., but they don't show fitness
l21: size scenarios usually are linked **to recurrence** periods. The de-coupling from size scenarios to recurrence period does not make sense.

1. **Introduction**
l26ff: what about debris flows, avalanches, shallow landslides, etc?
l29: at the border between China and Nepal.
l30: crosses/leads trhough mountainous areas.
l33: book cited incorrectly, plus: does it really make sense to cite a book for common knowledge such as "rockfalls usually occur in montaineous regions"?
L38: derived from a digitital elevation model
l41: This is not the only reason for LiDAR scanning and the Fanos et al. source is clearly focused on something different (machine learning for rockfall trajectory propagation modelling)
l42: The conclusion from the cited work is rather, that it is no unambiguous SAT derivation possible. That terrain is an important basis, is common knowledge. Not many rockfalls occur in the planes.
L44: rockfall susceptiblity is a combination of all of those factors. It should not be opposing, but complementary assessments.
L47: sentence makes no sense. Source areas can be identified more accurately either by using empirical, statistical or deterministic methods.
l49: how widely used is RHRS? And how accurate/universal are the proposed exponential function within the original RHRS publication? It is a method amongst many.
L51: Oommen at al. (1984) should be Bouali et al (2019)

L54-56: arguing with academic references from roughly 30 years ago, that a method is commonly used is a bit far fetched. The problematic on input data is already discussed there.

L62 ff: What is 3D collapse motion? The argument, that those models require extensive field investigation and experimental parameters as opposed to Flow-R is not substantial.

The reference Jabodeyoff et al.2003 is a link to where no manual for FLOW-R is found anymore (CONEFALL and others are found there). The statement, that FLOW-R produces more realistic results with the citations of a wrong manual is a bold – if not scientifically fraudulent - claim.

L 72: What is a fragment in this case? Usually fragments are fragmented parts from a initially released rock from the release area. Of course, those rocks are also fragments from the original rock wall etc, but in the literature, fragmentation means the breaking up of a single block during its trajectory. The influence on deposition patterns etc. is a hot topic and controversially debated. Reach angle analysis, however, can not contribute, to this discussion.

L 80ff: What is the temporal probability? Recurrence periods? There is a great many work around scenario building in rockfall etc. The authors have a point, that a thorough link between occurrence probability and scenario probability might be a weak point of current hazard mitigation literature. Please rephrase.

L85-90: Please refine the English.

2. **Study Area**

L93 Intence → Intense erosion and weathering processes

L94 600 m

Figure 1: all anticlines in the figure are labelled incorrectly → antivline

L101 The lithology in the area consists mainly of purplish-red mudstone

L105 Anticline → anticline

l104 frastures → infrastructure. Are there any statistics on the events on this road and the caused dammage?

L105 Anticline → anticline

L107: how obvious? What do you want to tell the reader?

L111: nucleus → core, near-wings?

L112: what is differential weathering?

L117: Figure 2a shows no vehicle damage, that is Figure 2d. "The sandstone cliff collapsed" are the steep section without vegetation, not necessarily a collapse already.

3. **Methodology**

Figure 3 – the presented methodology is a quite intricate interplay. A priori MLRM and RFM models work only for large data sets.

L144: SAT methodology according to Loye et al. shows quite a bit of DEM resolution dependency. The adaption of this procedure to a 10 m DEM is questionable.

L201: Reference should be Bak et al. (1988). Additionally, the reference deals with "Self-organized criticality"

L203 DOI of source Pelletier et al. 1997 is invalid.

[Figure]

Additionally, 27° is the transitions between footslopes and steep slopes. It is a rather low value in general as threshold.

l257: In general, the Varnes et al. (1984) citation is very old, hard to retrieve and in the context of MDLP highly likely the wrong citation.

L295: Comparison with SAT model approach is not valid, as SAT model approach is done incorrectly.

4.2 Temporal and size probability of rockfall sources

5. **Discussion**

l408ff: Are you altering Flow-R in order to incorporate all the promised things

---

## Author Comment (AC1)

**Point-by-point response to the reviews**

Dear Referee 1,

We would like to thank you for your professional and constructive comments concerning our manuscript entitled " Understanding rockfalls along the national road G318 in China: from source area identification to hazard probability simulation ". These comments are all valuable and helpful for revising and improving our manuscript. The main corrections in the manuscript and point-by -point responses are as following (the page number and line number in this refer to the revised manuscript)

**Technical corrections:**

General comment on the use of parentheses: Clearly a matter of writing style, however, IMHO the excessive use of parentheses hinders the reading flow. Personal guidance is: if it's important, rephrase it into the written sentences, if it does not merit being included in the text, remove it. The authors might check their use of parentheses with this in mind, or discard it as the referee's spleen. Does not hold for introduction of acronyms, of course.

Response: Thanks for your kind advice. In order to make it easier for readers to read, we have revised the extra parentheses in the paper.

Figure font sizes: Revise the font size and general sizing of heavily loaded figures.

Response: Thanks for your kind advice. We have modified the font size of all the figures.

**Abstract:**

l14: kinemics - kinematics, but is it really?

l20: results agree with measurements, fit well the acquired field data, etc., but they don't show fitness

l21: size scenarios usually are linked to recurrence periods. The de-coupling from size scenarios to recurrence period does not make sense.

Response: Thanks for your kind suggestion. In Line14, we modified the wrong words expressing kinemics to kinematics. The fitness is discussed from two aspects in this research: rockfall source area and trajectory probability. The fitness of rockfall source area is assessed in Table 4 and the comparison of different models is taken in Table 5.The fitness of trajectory is assessed in section 4.3. Fig.11 and Fig.12 shows the fitness by comparing historical and simulation results. It was indicated that the average difference value of run out is 1.97 m with an evaluation error ratio 3.66% (Line 372-373). Thank you for the suggestion that the fitness should be shown more detail in the abstract. The explanation about size scenario and return period scenarios can be found from the response "4.2 Temporal and size probability of rockfall sources"

**1. Introduction**

l26ff: what about debris flows, avalanches, shallow landslides, etc?

Response: Thanks for your kind suggestion. We revised the sentences as: Rockfall is one of the geological hazard…..

l29: at the border between China and Nepal. l30: crosses/leads through mountainous areas.

Response: Thanks for your kind suggestion. We revised the sentences as; …… and finally ending at the border between China and Nepal. More than 70 percent crosses through mountainous areas.

l33: book cited incorrectly, plus: does it really make sense to cite a book for common knowledge such as "rockfalls usually occur in montaineous regions"?

Response: Thanks for your kind suggestion. We have deleted the redundant expression in Line 34.

L38: derived from a digitital elevation model

Response: Thanks for your kind suggestion. We revised the sentences as: …in which slope gradient map derived from digital elevation model is used.

l41: This is not the only reason for LiDAR scanning and the Fanos et al. source is clearly focused on something different (machine learning for rockfall trajectory propagation modelling)

Response: Thanks for your kind suggestion. We deleted this sentence.

l42: The conclusion from the cited work is rather, that it is no unambiguous SAT derivation possible. That terrain is an important basis, is common knowledge. Not many rockfalls occur in the planes.

Response: Thanks for your kind suggestions. We corrected inaccurate statement as: SAT value is an important basis for Rockfall hazard Assessment.

L44: rockfall susceptibility is a combination of all of those factors. It should not be opposing, but complementary assessments.

Response: Thanks for your kind suggestion. We have revised the sentences as:

The morphology-based method is simple in data-limited areas. If data available or assessment scale is large, other conditioning factors such as discontinuities and joint sets in rocks need to be supplemented (Guzzetti et al., 1998; Jaboyedoff et al., 2003; Frattini et al., 2008; Heckmann et al., 2016).

L47: sentence makes no sense. Source areas can be identified more accurately either by using empirical, statistical or deterministic methods.

Response: Thanks for your kind suggestions. We have corrected the sentence as:

Source areas can be identified more accurately either by using empirical, statistical or deterministic methods.

l49: how widely used is RHRS? And how accurate/universal are the proposed exponential function within the original RHRS publication? It is a method amongst many.

Response: RHRS method has been used by many researchers, such as Brawner et al. (1975); Pierson (1993); Budetta (2004); Li et al. (2009); Corominas et al. (2013). The Rockfall Hazard Rating System (RHRS) is a stepwise process designed to identify potentially hazardous slopes by assigning a hazard rating. In Line 49, we did not mention exponential function within the original RHRS publication. If more explanation is needed, please don't hesitate to tell the authors.

L51: Oommen at al. (1984) should be Bouali et al (2019)

Response: Thanks for your kind suggestions. We corrected the incorrect citations by changing Oommen at al. (1984) to Bouali et al. (2017).

Bouali, E. H., Oommen, T., Vitton, S., Escobar-Wolf, R., Brooks, C.: Rockfall Hazard Rating System: Benefits of Utilizing Remote Sensing. Environmental and Engineering Geoscience. 23 (3), 165–177. https://doi.org/10.2113/gseegeosci.23.3.165, 2017.

L54-56: arguing with academic references from roughly 30 years ago, that a method is commonly used is a bit far fetched. The problematic on input data is already discussed there.

Response: Thanks for your kind suggestion. We have revised the references to Lee, 2005; Benchelha et al., 2019. The sentence is revised as:

MLRM is used to construct slope instability susceptibility models (Chung et al., 1995; Lee, 2005; Benchelha et al., 2019).

Lee S. Application and Cross-Validation of Spatial Logistic Multiple Regression for Landslide Susceptibility Analysis. Geosciences Journal, 9(1):63-71. https://link.springer.com/content/pdf/10.1007/BF02910555.pdf, 2005.

Benchelha, S., Chennaoui Aoudjehane, H., Hakdaoui, M., El Hamdouni, R., Mansouri, H., Benchelha, T., Layelmam, M., Alaoui, M., Landslide susceptibility mapping in the Commune of Oudka, Taounate Province, north Morocco; A comparative analysis of logistic regression, multivariate adaptive regression spline, and artificial neural network models. Environ. Eng. Geosci. 26(2), 185-200. https://doi.org/10.2113/EEG-2243, 2019.

L62 ff: What is 3D collapse motion? The argument, that those models require extensive field investigation and experimental parameters as opposed to Flow-R is not substantial.

The reference Jabodeyoff et al.2003 is a link to where no manual for FLOW-R is found anymore (CONEFALL and others are found there). The statement, that FLOW-R produces more realistic results with the citations of a wrong manual is a bold – if not scientifically fraudulent - claim.

Response: Thanks for your kind suggestion. The expression of 3D Rockfall Motion is not accurate, so we removed it in Line 78. It means that the three-dimensional physical model is used to evaluate the risk of rockfall. The argument that those models require extensive field investigation and experimental parameters as opposed to Flow-R is deleted. The sentence is revised as:

Among them, Flow-R is developed for regional-scale on Matlab@2016, utilizing both empirical studies and physical modeling for gravitational hazards (Horton et al., 2013).

In Line 81, the sentence we expressed was not exact, so we deleted it to avoid ambiguity. The statementthat FLOW-R produces more realistic results with the citations of a wrong manual is deleted.

L 72: What is a fragment in this case? Usually fragments are fragmented parts from a initially released rock from the release area. Of course, those rocks are also fragments from the original rock wall etc, but in the literature, fragmentation means the breaking up of a single block during its trajectory. The influence on deposition patterns etc. is a hot topic and controversially debated. Reach angle analysis, however, can not contribute, to this discussion.

Response: Thanks for your kind suggestions. The expression of 'fragment' is not accurate. In order to avoid ambiguity, we revised this sentence as:

In trajectory path simulation, the minimum reach angle or shadow angle is a key parameter controlling the influence area of rockfall.

L 80ff: What is the temporal probability? Recurrence periods? There is a great many work around scenario building in rockfall etc. The authors have a point, that a thorough link between occurrence probability and scenario probability might be a weak point of current hazard mitigation literature. Please

rephrase.

Response: Thanks for your kind suggestions. Temporal probability refers to the probability of n disasters occurring within a certain period T in a certain region. More explanation can also be found in the responses for you "4.2 Temporal and size probability of rockfall sources".
"

L85-90: Please refine the English.

Response: Thanks for your kind suggestions. We reworked the English expression to make it easier for readers to understand.we revised the sentences as:

For quantitative risk assessment of rock fall hazard, we consider that the rockfall assessment including source area identification and rock fall propagation should be at the level of probability assessment. To understand the potential risk from rock falls along national highway G318 in China, this study try to assess the potential hazard probability and risky road segments, considering a given rock fall volume over certain return period.

**2. Study Area**

L93 Intence → Intense erosion and weathering processes

L94 600 m

Response: Thanks for your kind suggestion. We corrected the wrong words.

Figure 1: all anticlines in the figure are labelled incorrectly antivline

Response: Thanks for your kind suggestions. We have modified all the wrong words in the picture, as shown below:

[Figure]

L101 The lithology in the area consists mainly of purplish-red mudstone

l104 frastures    infrastructure. Are there any statistics on the events on this road and the caused dammage?

L105 Anticline    anticline

Response: Thanks for your kind suggestion. We have modified the wrong words. There are no statistical data of the events on this road but we have known in the field that there was a hazard damage to a pick-up car with 3 people inside six years ago (Fig.2d).

L107: how obvious? What do you want to tell the reader?

Response: Thanks for your kind suggestion. Sorry for the unclear and confusing statement. We have deleted the sentence.

L111: nucleus    core, near-wings?

Response: Thanks for your kind suggestion. We modified the wrong word in Line 147.

L112: what is differential weathering?

Response: The effects of different weathering of lithology of cliffs or steep slopes are the cause for rockfall. Soft rocks like mudstone are easier or slower to be weathered than hard stones like sandstone. If the slope is composed by mudstone overlaid by sandstone, differential weathering will cause caves below cliffs and then rockfalls.

L117: Figure 2a shows no vehicle damage, that is Figure 2d. "The sandstone cliff collapsed" are the steep section without vegetation, not necessarily a collapse already.

Response: Thanks for your kind suggestion and sorry for the confusing labels in Figure 2. We have modified the figures and labels very carefully.

[Figure]

**4. Methodology**

Figure 3 – the presented methodology is a quite intricate interplay. A priori MLRM and RFM models work only for large data sets.

Response: Thanks for your kind suggestions. Our study area is about 21 km², with 108 rockfall source areas investigated in the study area. The data is sufficient for the use of MLRM and RFM models.

L144: SAT methodology according to Loye et al. shows quite a bit of DEM resolution dependency. The adaption of this procedure to a 10 m DEM is questionable.

Response: Thanks for your kind suggestion. According to Figure 2 by Loye et al.(2009), there is a suggested grid size from the relationship between grid size, slope angle and slope height. As to our study area, the average slope angle is about 40 (Fig.5) and the major slope height over 5 m. The grid size 10 m is no questionable for this study.

L201: Reference should be Bak et al. (1988). Additionally, the reference deals with "Self-organized criticality"

Response: Thanks for your kind suggestion. We have corrected the reference.

L203 DOI of source Pelletier et al. 1997 is invalid.

Response: Thanks for your kind suggestions. We have modified DOI in Pelletier et al. 1997.

[Figure]

Additionally, 27° is the transitions between footslopes and steep slopes. It is a rather low value in general as threshold.

Response: Thanks for your kind suggestions. The SAT we used is different from the method of considering only topographic slope data in reference Loye et al. (2009). We not only considered the topographic slope in the study area, but also historical disaster rockfall data in the study area. The relationship between slope angle and historical rockfalls in the study area is shown in the figure below. It can be seen from the green line in the following figure that the regional distribution proportion of rockfall above 27° is higher than that in Loye et al. (2009). So, 27° is a suitable threshold in our study area where the terrain is steep.

[Figure]

l257: In general, the Varnes et al. (1984) citation is very old, hard to retrieve and in the context of MDLP highly likely the wrong citation.

Response: Thanks for your kind suggestion. We re-checked the literature and modified the references, changing Varnes et al. (1984) to Rissanen (1978) and Vitanyi (2000).

Rissanen, J. J.: Modeling by the shortest data description, Automatica-J.IF AC, 14, 465–471, https://doi.org/10.1016/0005-1098(78)90005-5, 1978.

Vitanyi, P. M. B., Li, M.: Minimum description length induction, Bayesianism, and Kolmogorov complexity, IEEE Transactions on Information Theory, 46, no. 2, 446-464, https://doi.org/10.1109/18.825807 , 2000.

L295: Comparison with SAT model approach is not valid, as SAT model approach is done incorrectly.

Response: Thanks for your kind suggestion. According to the method by Loye et al. (2009), we made the Gaussian distribution graph of terrain slope in the study area, as shown below. It can be seen that 33° should be selected according to the way of considering only terrain. However, according to the statistical historical rockfall data, only 57.85% of the rockfall disaster is located above 33°, while 84.79% of the rockfall disaster is located above 27°. If 27° is selected, a large amount of historical data is retained, which provides a guarantee for the accurate prediction of the rockfall source area later.

[Figure]

**4.2 Temporal and size probability of rockfall sources**

Response: Thanks for your kind suggestions. Combining temporal, spatial and size probability to assess hazard probability is proved to be a scientific way according to the published papers, such as:

Wu, C.Y., Chen, S.C.: Integrating spatial, temporal, and size probabilities for the annual landslide hazard maps in the Shihmen watershed, Taiwan. Nat. Hazard. Earth Sys. 13(9), 2353-2367. https://doi.org/10.5194/nhess-13-2353-2013, 2013.

Guzzetti, F., Galli, M., Reichenbach, P., Ardizzone, F., Cardinali, M.: Landslide hazard assessment in the Collazzone area, Umbria, central Italy. Nat. Hazard. Earth Sys. 6(1), 115-131. https://doi.org/1684-9981/nhess/2006-6-115, 2006.

Catani, F., Casagli, N., Ermini, L., Righini, G., Menduni, G.: Landslide hazard and risk mapping at catchment scale in the Arno River basin. Landslides. 2(4), 329-342. https://doi.org/10.1007/s10346-005-0021-0, 2005.

Brunetti, M.T., Guzzetti, F., Rossi, M.: Probability distributions of landslide volumes. Nonlinear Proc. Geoph. 16(2), 179-188. https://doi.org/10.5194/npg-16-179-2009, 2009.

Liu, B., Siu, Y.L., Mitchell, G., Xu, W.: Exceedance probability of multiple natural hazards: Risk assessment in China's Yangtze River Delta. Nat. Hazards. 69(3), 2039-2055. https://doi.org/10.1007/s11069-013-0794-8, 2013.

Melchiorre, C., Frattini, P.: Modelling probability of rainfall-induced shallow landslides in a changing climate, Otta, Central Norway. Climatic Change. 113(2), 413-436. https://doi.org/10.1007/s10584-011-0325-0, 2012.

Fu, S., Chen, L., Woldai, T., Yin, K., Gui, L., Li, D., Du, J., Zhou, C., Xu, Y., Lian, Z.: Landslide hazard probability and risk assessment at the community level: A case of western Hubei, China. Nat. Hazard. Earth Sys. 20(2), 581-601. https://doi.org/10.5194/nhess-20-581-2020, 2020.

In this research, we aims to find the quantitative probability in terms of rockfall occurrence frequency, travel distance and size, which relate to quantitative risk assessment of elements at risk. Also it is requested by local government for future management in the study area. We collected sufficient historical rockfall hazard and geological environmental data in the field. Poisson model and exponential equation are then applied to obtain temporal and size probability. Considering the risk management request, 5, 20 and 50 years return periods are requested scenarios for hazard prevention budget plan of local government. It is why we use these three return periods to assess the potential risk.

**5. Discussion**

l408ff: Are you altering Flow-R in order to incorporate all the promised things

Response: Yes, We have made the plan to improve the algorithm in Flow-R for better understanding the rock fall risk in our study area.

We tried our best to improve the manuscript. We feel great thanks for your professional review work on our manuscript, and hope that the correction and response will meet with approval.

Sincerely,

Lixia Chen

---

## Author Comment (AC2)

Dear Referee 2,

Thank you very much for your professional comments on our manuscript. These comments are all valuable and helpful for revising and improving our manuscript. The main corrections in the manuscript and the point-by-point responses to your comments are as following (the page number and line number in this letter refer to the revised manuscript):

This paper is the application of several existing methodologies to a given case study. I don't recommend publication of this paper for the following reasons: 1) it does not fit the requirements for the scientific paper since I did not identified scientific novelty, 2) the relevance of the complete approach is questionable.

Response: Thank you for your comments. 1) This case study is taken based on the previous studies. Meanwhile, we created an improved way to minimize the uncertainty of source area susceptibility both considering slope angle and important controlling factors of rock falls. It is proved by the study that the potential source area grids reduced from 160,823 to 4,002 with only 1.4 percent loss of historical rock fall samples. The simulation efficiency increased about 40 times, which highly reduced the burden of trajectory simulation. 2) Detailed field investigation was taken by the authors to understand the mechanism of rock fall and related risk along the road. Also, the methodology is based on previous public approaches and applied for the study area where rock fall hazard is an important risk source. The local government needs quantitative risk assessment result to guide their management work. If detailed explanation is needed, please refer to our response to the Referee 1.

Sincerely,
Lixia Chen

---

## Author Comment (AC3)

Dear Referee 3,

Thank you very much for your professional comments on our manuscript. These comments are all valuable and helpful for revising and improving our manuscript. The main corrections in the manuscript and the point-by-point responses to your comments are as following (the page number and line number in this letter refer to the revised manuscript):

This paper presents a concoction of standard methods in rockfall hazard analysis and does not contain a substantial contribution to science. The reviewer can concur with the recommendations of RC2 and the assessment of RC1.

Response: Thank you for your comments. Please refer to our response to the Referee 1 and 2. If more detailed explanation is needed, please don't hesitate to tell the authors.

Sincerely,
Lixia Chen